# Systematic Review with Trial Sequential Analysis of Prophylactic Antibiotics for Acute Pancreatitis

**DOI:** 10.3390/antibiotics11091191

**Published:** 2022-09-03

**Authors:** Goran Poropat, Karla Goričanec, Alojzije Lacković, Andrea Kresović, Antun Lončarić, Martina Marušić

**Affiliations:** 1Department of Gastroenterology, Clinical Hospital Center Rijeka, Faculty of Medicine, University of Rijeka, 51000 Rijeka, Croatia; 2Faculty of Medicine, University of Rijeka, 51000 Rijeka, Croatia; 3Department of Cardiology, General Hospital ‘‘Dr. Ivo Pedisic’’ Sisak, 44000 Sisak, Croatia; 4Department of Emergency Medicine, General Hospital Zadar, 23000 Zadar, Croatia

**Keywords:** anti-bacterial agents, meta-analysis, pancreatic necrosis, randomized controlled trials, sepsis

## Abstract

Background/Objectives: Prophylactic antibiotics (PAB) are being still widely used for treatment of acute pancreatitis (AP) despite trials showing no firm evidence of efficacy. We aimed to evaluate effects of PAB for AP in a meta-analysis and the need for further research by trial sequential analysis (TSA). Methods: Medline, Scopus and Web of Science were searched for randomized clinical trials. Primary outcomes were all infections and mortality. Secondary outcomes comprised infected pancreatic necrosis (IPN), specific infections, organ failure, surgical interventions, and length of hospital stay. Results: Twenty-one trials with 1383 pts were included. PAB were received by 703 pts, while 680 were controls. Mortality was similar with RR 0.85 (95% CI 0.66–1.10). Infections were significantly reduced (RR 0.60; 95% CI 0.49–0.74), mainly due to decreased risk of sepsis (RR 0.43; 95% CI 0.25–0.73) and urinary tract infections (RR 0.46; 95% CI 0.25–0.86). No significant reduction for IPN was shown (RR 0.81; 95% CI 0.63–1.04). Length of hospital stay was diminished by MD −6.65 (95% CI −8.86 to −4.43) days. TSA for all infections showed that the cumulative Z score crossed both conventional and monitoring boundaries at 526 pts from a heterogeneity-corrected required information size of 1113 pts based on a 40% incidence of infections in the control group, RRR of 30%, alpha 5%, beta 20%, and heterogeneity 56%. Conclusions: PABs decrease the rate of infections in AP, mainly due to RRR of extra-pancreatic infections, requiring no further research. No significant effect is shown on IPN and mortality, although firmer evidence is needed.

## 1. Introduction

Acute pancreatitis (AP) is the most common acute hospital inflammatory condition in gastroenterology without a known and established pathogenetic treatment. With an incidence of about 29–44 cases per 100,000 person-years [1], approximately 20% of patients develop a necrotizing form of disease or organ failure with a moderately severe or severe clinical course, resulting in a mortality rate of 20–40% [2]. In its early course, AP may be characterized by organ failure induced by an intense systemic inflammatory response. Infected pancreatic necrosis and sepsis are more commonly present in the later phases, possibly leading to late-onset organ failure. However, infectious complications may also occur in its early phase [3], making it difficult to determine the cause of organ failure.

Existing evidence from randomized controlled trials [4,5,6] and meta-analyses [7,8,9] have suggested that prophylactic antibiotic use has no part in the treatment strategy for patients with acute pancreatitis. Despite such recommendations being advocated by different guidelines and societies [10,11,12,13], worldwide compliance to these recommendations is still poor, mostly due to fear from infectious complications which may lead to devastating outcomes [14]. This approach has led not only to increased treatment costs, but is also associated with potentially harmful effects by increasing development of multi-drug resistant bacteria and fungal infections [14,15]. Prior research focused mainly on reduction of infected pancreatic necrosis as a factor significantly affecting clinical outcome by raising morbidity and mortality. As a relatively uncommon complication of AP, most of these trials have been substantially underpowered. Furthermore, the role of extra-pancreatic infections and their influence on the course of the disease, as well as how prophylactic antibiotics affect development of these potentially life-threatening complications has not been investigated enough and is still rather unclear. We aimed to perform a comprehensive systematic review with meta-analysis of randomized controlled trials assessing the prophylactic use of antibiotics in patients with AP by evaluating all clinically relevant outcomes. Furthermore, to define the required information size for specific outcomes and to assess the risk of random error within the meta-analyses, we decided to perform trial sequential analysis (TSA), a methodology that combines the calculated sample size for a meta-analysis with the threshold of statistical significance [16,17]. 

## 2. Materials and Methods

This systematic review of randomized controlled trials assessing the prophylactic use of antibiotics in acute pancreatitis was conducted and reported according to the PRISMA guidelines [18]. The protocol is registered and available in the PROSPERO registry (CRD42018105977). 

### 2.1. Literature Search

Two authors independently searched for randomized controlled trials in Medline, Web of Science, Scopus, and The Cochrane Library (CENTRAL) from inception until February 2022. Detailed search strategies for all electronic databases are given in Appendix B. Reference lists of identified trials were additionally hand-searched, and corresponding authors were contacted if further information were required. 

### 2.2. Selection Criteria

We included only randomized controlled trials assessing the prophylactic use of antibiotics compared to placebo, no intervention, or any other intervention in patients with acute pancreatitis, regardless of language and publications status, etiology of pancreatitis, type of antibiotic used, dosage, route of administration, timing of initiation, and duration of treatment. 

### 2.3. Data Extraction and Analysis

Primary outcome measures comprised any infection and mortality. Secondary outcome measures included specific complications (infected pancreatic necrosis, sepsis, bacteremia, pneumonia, urinary tract infection (UTI), and other infections), organ failure, length of hospital stay, and need for surgical intervention.

Two authors extracted and validated information independently from selected trials by using data extraction forms specifically designed for this purpose. The following information were retrieved: primary author, country of origin, trial design, number of participants allocated to study groups, age and gender, etiology of AP, intervention regimens provided, and period of follow-up. 

All statistical analyses were performed using Review Manager 5.2. Results for dichotomous outcomes were expressed as risk rations (RRs) with 95% confidence intervals (CIs), while for continuous outcomes as mean differences (MDs) with 95% CIs were used. Heterogeneity was assessed by means of the I2 value, and was defined as low (<25%), moderate (25 to 50%), or high (>50%). Meta-analysis was performed by using the fixed-effect model in cases of low heterogeneity, otherwise both the fixed-effect and random effects models were used. When significant differences in results obtained by the two models were present, results from both models were reported. If no differences were observed, only results from the fixed-effect model were shown.

### 2.4. Quality Assessment

Methodological quality was assessed by two authors independently using the Cochrane tool for assessment of risk of bias in RCTs. The following domains were assessed: generation of the allocation sequence, allocation concealment, blinding, incomplete outcome data, selective outcome reporting, and other sources of bias. Any potential disparities between authors were resolved by discussion or by consultation of a third review author to arbitrate the decision.

### 2.5. Trial Sequential Analysis

Trial sequential analysis was performed for primary outcomes and secondary outcomes showing a statistically significant result in meta-analysis. Trial sequential monitoring boundaries were constructed based on the heterogeneity-corrected required information size (HCRIS) with an alpha of 5%, a beta of 20%, and the corresponding heterogeneity. We estimated a relative risk reduction (RRR) in the intervention group of 30% assuming the proportion of participants in the control group with the outcome of interest according to data from previous published trials and reports.

## 3. Results

A total of 5170 references were retrieved by searching electronic databases. After erasing duplicates 4901 references remained. Twenty-eight articles were selected after screening through titles and abstracts. We selected 21 articles for inclusion according to our criteria, while 7 were excluded. A total of 21 studies with 1383 randomized patients were finally included in the analysis (Figure 1). Prophylactic antibiotics were received by 703 patients, while 680 were assessed as controls. Characteristics of included studies are shown in Appendix A. 

Nine studies with a total of 526 randomized patients reported on all infectious complications (Figure 2), and all included studies reported on mortality (Figure 3).

The analyses of specific infections showed that PAB significantly reduced sepsis (17/255 vs. 32/238; RR 0.43; 95% CI 0.25 to 0.73; I^2^ = 0%) and UTIs (17/254 vs. 33/258; RR 0.46; 95% CI 0.25 to 0.86; I^2^ = 11%). No significant difference was detected between the groups regarding other specific infections including pneumonia (29/254 vs. 40/258; RR 0.73; 95% CI 0.49 to 1.09; I^2^ = 0%), infected pancreatic necrosis (86/561 vs. 101/541; RR 0.81; 95% CI 0.63 to 1.04; I^2^ = 0%), bacteremia (9/81 vs. 9/78; RR 0.92; 95% CI 0.33 to 2.58; I^2^ = 0%), and other infections (22/237 vs. 24/241; RR 0.90; 95% CI 0.49 to 1.66; I^2^ = 4%). Fungal infections as potential adverse events of PAB treatment were assessed in 9 trials including 534 patients. No significant differences were shown (16/260 vs. 20/274; RR 0.84; 95% CI 0.43 to 1.61); I^2^ = 20%).

The rate of organ failure was similar in both groups. This analysis was based on eight trials with 552 randomized patients (79/277 vs. 95/275; RR 0.82; 95% CI 0.65 to 1.03; I^2^ = 0%). There were also no significant differences when specific organ failures were assessed individually. Acute renal failure occurred in 30/214 vs. 36/212 patients (RR 0.78; 95% CI 0.46 to 1.35; I^2^ = 0%); acute respiratory failure occurred in 62/214 vs. 72/212 patients (RR 0.77; 95% CI 0.50 to 1.18; I^2^ = 0%); and cardiovascular failure occurred in 13/149 vs. 16/148 patients (RR 0.78; 95% CI 0.36 to 1.70; I^2^ = 0%). The need for surgical interventions was assessed in 15 trials and showed no significant difference (111/525 vs. 128/510; RR 0.79; 95% CI 0.58 to 1.07; I^2^ = 0%). The length of hospitalization was significantly decreased in the PAB group (Figure 4).

### 3.1. Trial Sequential Analysis

TSA according to the protocol was performed for the following outcomes: any infectious complications, mortality, sepsis, UTI, and length of hospital stay. Detailed graphs and descriptions are presented in Figure 5, Figure 6, Figure 7, Figure 8 and Figure 9.

### 3.2. Assessment of Risk of Bias

Overall, only the trial by Poropat et al. was assessed as having a low risk of bias. Random sequence generation was judged as low risk of bias in five trials (24%), while allocation concealment was judged as low risk in four trials (19%). Three trials performed adequate blinding of participants, personnel, and outcome assessors (14%). Thirteen trials had low risk of attrition bias (62%), and twelve had low risk of selection bias (57%). No other potential sources of bias were detected in 14 trials (67%). Detailed risk of bias assessment is given in Appendix A.

## 4. Discussion

This systematic review with meta-analysis and TSA examined the evidence from 21 randomized controlled trials with a total of 1383 participants for the use of antibiotic prophylaxis in acute pancreatitis. The results suggest that antibiotic prophylaxis may reduce the incidence of infectious complications, most likely due to the prevention of extra-pancreatic infections. Nevertheless, there is insufficient evidence to assess the impact of treatment on overall clinical outcomes.

The role of empiric antibiotics in preventing septic complications of acute pancreatitis has been controversial for over five decades. The results of early meta-analyses indicated a beneficial effect of antibiotic prophylaxis, reflected in a statistically significant decrease in mortality [19,20,21,22], pancreatic infections [21,22], and sepsis [20,22]. However, the initiation of large randomized, double-blind, placebo-controlled trials [3,4,5,6,23] did not confirm the previously recognized utility of antibiotic therapy. Consequently, antimicrobials are only indicated when an infection is either confirmed or highly suspected [10,11,12,13], as recent evidence from RCTs has been affirmed by contemporary systematic reviews [8,24,25,26,27]. Consistent with most recent studies, our meta-analysis found that antibiotic prophylaxis had no statistically significant effect on mortality. An explanation for the inconsistency of the meta-analytic results was proposed by De Vries et al. [27], who discovered an inverse correlation between the methodological quality of RCTs and the survival benefit of early administration of antibiotics. However, the conduction of TSA unveiled that the number of participants was insufficient to reliably assess the assumed intervention effect of 30% relative risk reduction. Accordingly, there is no firm evidence on the impact of prophylactic treatment on mortality. Consequently, the application of conventional significance thresholds (95% CI and *p*-value of 0.05) in the statistical analysis may lead to spurious results. Conventional statistical intervals do not consider the amount of data available relative to the required information size [28]. Therefore, the reliability of statistical significance is often overestimated, especially when data are sparse, leading to false-positive (type I error) or false-negative results (type II error) [28]. The risk of type I error is further enhanced when significance tests are derived repeatedly, such as when meta-analyses are updated while accumulating data from additional studies [17]. Furthermore, in the absence of statistical significance, conventional statistical thresholds cannot distinguish between ineffective interventions and lack of evidence due to an underpowered meta-analysis. As a result, an effective intervention may be erroneously rejected [28].

Furthermore, the findings from the present meta-analysis indicated that the cumulative incidence of infectious complications was significantly diminished in the group receiving antibiotic prophylaxis compared with the control group. Adjustment of the significance intervals with trial sequential monitoring boundaries declared a sufficient level of evidence to support a beneficial treatment effect. In addition, the meta-analysis showed that the risk of UTIs and sepsis was significantly reduced when prophylaxis was applied. Therefore, it is likely that prophylactic antibiotics reduce the overall incidence of infectious complications, with greater mitigation of UTIs and sepsis than pancreatic necrosis infection. These observations are consistent with the recent review of 11 RCTs [26] in which the authors reported a decrease in extra-pancreatic infections (particularly UTIs) with antibiotic prophylaxis. Even so, the TSA for UTIs and sepsis signified a lack of reliable evidence for conclusive results. Although recent discoveries refer that antibiotics may play a protective role in non-pancreatic infections, the relevance of these infections to clinical outcomes remains to be determined. While some authors have found no impact of extra-pancreatic infectious complications on the course of acute pancreatitis [29], others associate them with a higher risk of organ failure [30], intensive care unit admission, and higher APACHE II scores [31]. In addition, there is evidence that both bacteremia and lung infections correlate with a higher incidence of pancreatic necrosis infection and death [32]. The data from the present meta-analysis considering these specific extra-pancreatic infections did not reveal notable differences between groups comparing prophylactic antibiotics with controls. However, a prophylactic strategy was associated with a statistically significant reduction in the length of hospital stay. Here, the result of the meta-analysis corresponded with the findings of the TSA, indicating that no further evidence is needed. In contrast to several publications [7,15,33,34] suggesting that prior antibiotic treatment may contribute to the development of infections with invasive fungal pathogens, the current analysis showed no difference in fungal infection rates with prophylactic antibiotics. There was also no significant correlation between prophylactic antibiotic management and other secondary outcomes.

The major strength of our review is that it introduces TSA into the statistical methods for evaluating antibiotic prophylaxis in acute pancreatitis. The trial sequential monitoring boundaries create thresholds based on the number of events available and the impact of repeated testing [17]. When the accumulated data are scarcer than the required information size, the monitoring boundaries have more stringent significance thresholds, reducing the likelihood of misinterpretation of random error [28]. In addition, the TSA establishes futility bounds based on a predefined minimum anticipated intervention effect to validate when the hypothesized outcome might be considered unattainable [17,35].

In addition, this comprehensive systematic review included the largest number of studies and participants on this subject to date but was nevertheless limited by an insufficient information size for precise effect estimates. Current meta-analyses of mortality, sepsis, and UTI are underpowered. As such, the results of the conventional statistical evaluation are prone to random error. The results from the TSA could guide trial investigators in planning and generating future studies, thus potentiating sufficiently powered meta-analyses [28]. To obtain an adequate information size for the expected intervention effect of 30% relative risk reduction, subsequent studies should enroll an additional 1638 participants for the mortality analysis, 1291 participants for the sepsis analysis, and 871 participants for the UTI analysis. Possibly, these numbers could be lower if trial sequential monitoring boundaries or futility thresholds were exceeded. Because the focus of previous studies has been on the prevention of pancreatic infection, CT-proven pancreatic necrosis was the inclusion criterion for most RCTs. However, different study designs and eligibility criteria would potentially allow for greater evidence in relation to non-pancreatic infections. Another limitation of the present review is the high prevalence of risk of systematic error (bias) among the included trials. Only one of the studies reviewed was classified as low risk, while the other 20 were at high risk of bias. Interestingly, the lack of an adequate blinding method [36,37,38,39,40,41] mainly resulted in reporting the beneficial effect of prophylactic antibiotics on septic complications and clinical outcomes. In contrast, a study with a low risk of bias [6] revealed no significant preference for routine antibiotic administration. Although the risk of systematic error in this study is low, the sample size of 101 patients does not provide sufficient data to draw decisive conclusions because of the risk of random error.

Our meta-analysis demonstrated some beneficial effects of antibiotic prophylaxis, manifested by reductions in length of hospitalization, overall infection rates, and extra-pancreatic infections, but without significant repercussions on mortality. Due to the lack of substantial evidence of appreciable treatment outcomes and the high risk of bias in the included studies, we cannot determine conclusive judgments about the overall efficacy of prophylactic antibiotics in acute pancreatitis. Additional high-quality clinical research is therefore inevitable. Further studies should also clarify the role of extra-pancreatic infections in the disease course and infer the risk of fungal infections as an adverse event of antibiotic prophylaxis. In addition, conducting TSA for other clinical questions could further elucidate the strength of existing evidence concerning this topic.

## Figures and Tables

**Figure 1 antibiotics-11-01191-f001:**
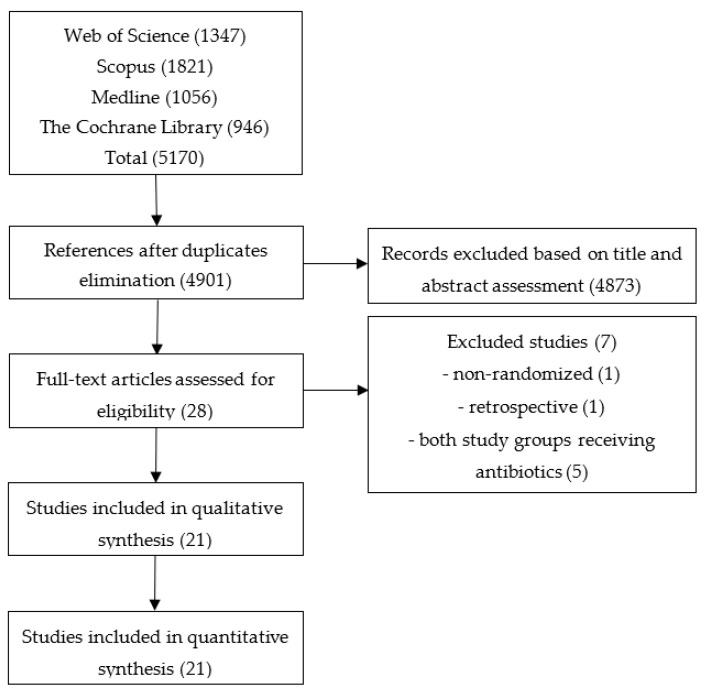
Flowchart illustrating the literature search and study selection.

**Figure 2 antibiotics-11-01191-f002:**
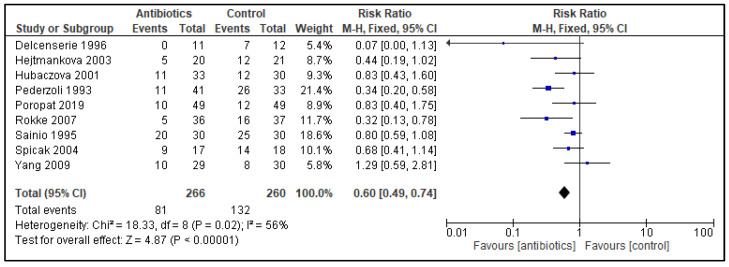
Forest plot of comparison of antibiotics versus control; outcome: all infectious complications. Infections were registered in 81 out of 266 patients receiving PAB versus a 132 out of 260 control patients, which resulted in a RR 0.60 (95% CI 0.49 to 0.74) favoring antibiotic usage, with a heterogeneity of 56%.

**Figure 3 antibiotics-11-01191-f003:**
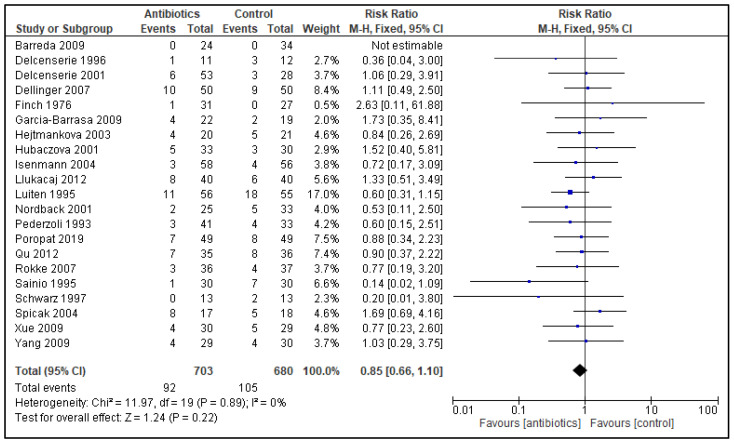
Forest plot of comparison of antibiotics versus control; outcome: mortality. Data from 21 trials showed 92 deaths occurring in the PAB group (N = 703) compared to the 105 deaths in the control group (N = 680). A RR of 0.85 (95% CI 0.66 to 1.10) and a heterogeneity of 0% confirmed no significant difference in mortality rates.

**Figure 4 antibiotics-11-01191-f004:**
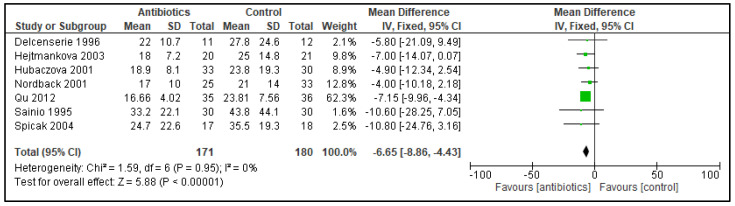
Forest plot of comparison of antibiotics versus control; outcome: length of hospitalization. These data were available from seven trials with a total of 351 patients (171 vs. 180), reporting a significant shortening of the length of hospitalization, by MD −6.65 days (95% CI −8.86 to −4.43) and a heterogeneity of 0%.

**Figure 5 antibiotics-11-01191-f005:**
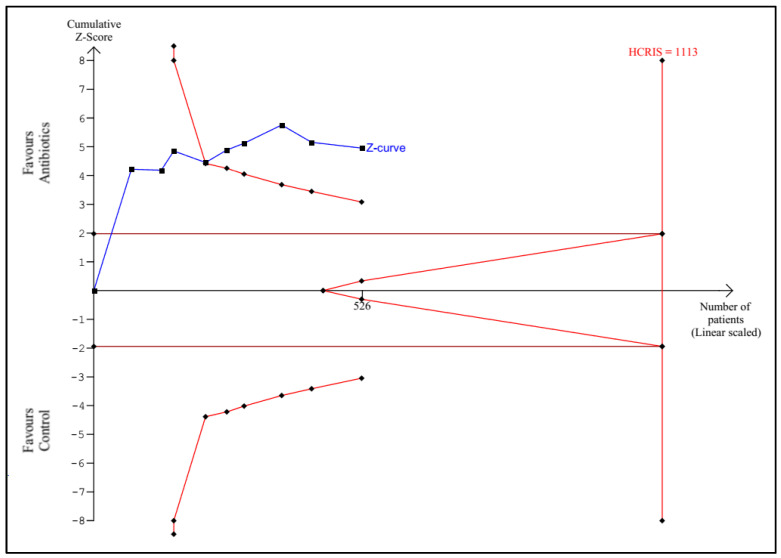
Trial sequential analysis for all infectious complications. The Z-curve crossed both the conventional and constructed monitoring boundaries at a total of 526 randomized patients (47% of the HCRIS) with an HCRIS of 1113 patients calculated based on a 40% incidence of infections in the control group, an RRR of 30%, an alpha of 5%, a beta of 20%, and heterogeneity of 56%; HCRIS = heterogeneity-corrected required information size.

**Figure 6 antibiotics-11-01191-f006:**
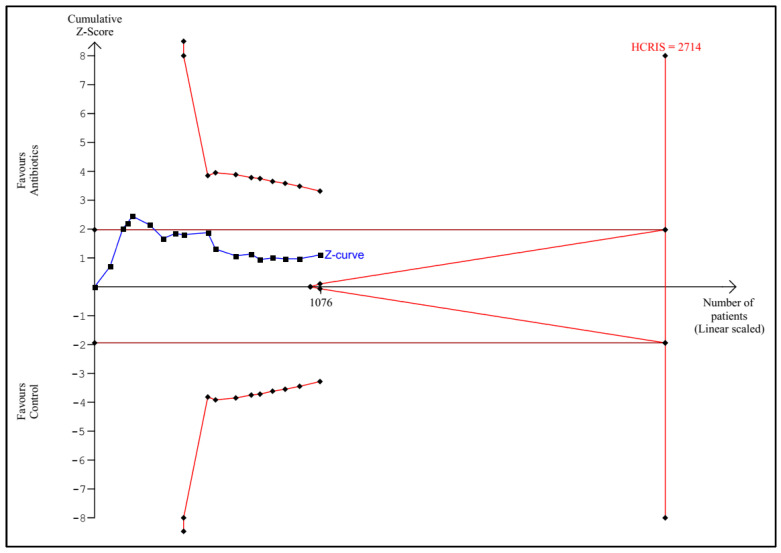
Trial sequential analysis for mortality. The cumulative Z-curve did not cross the conventional, nor the monitoring boundaries. The calculated HCRIS is 2714 patients based on an estimated mortality of 10% in the control group, a RRR of 30%, an alpha of 5%, a beta of 20%, and a heterogeneity of 0%. A total of 1076 patients have been randomized (40% of the HCRIS); HCRIS = heterogeneity-corrected required information size.

**Figure 7 antibiotics-11-01191-f007:**
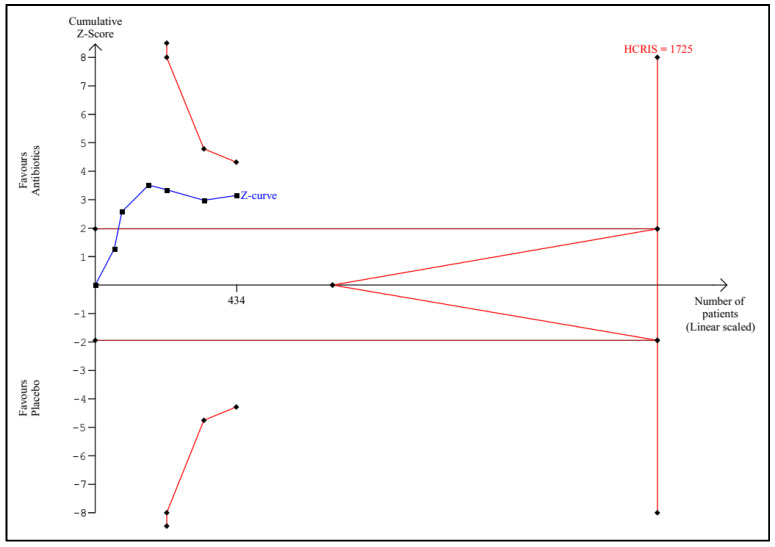
Trial sequential analysis for sepsis. The calculated HCRIS is 1725 patients. A total of 434 patients were randomized (25% of the HCRIS). The Z-curve crossed the conventional boundaries for *p* = 0.05, but not the monitoring boundaries. The calculation was performed with a heterogeneity of 0%, proportion of patients with sepsis in the control group of 15%, an RRR of 30%, an alpha of 5%, and a beta of 20%; HCRIS = heterogeneity-corrected required information size.

**Figure 8 antibiotics-11-01191-f008:**
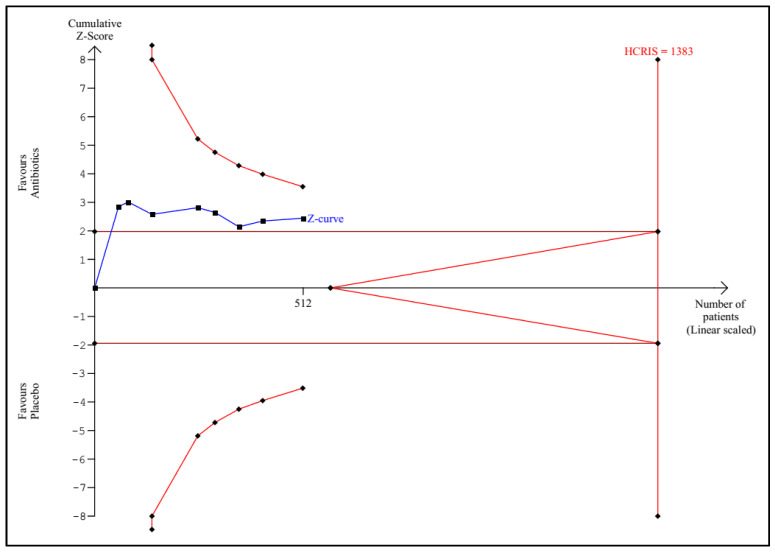
Trial sequential analysis for urinary tract infection. The HCRIS of 1383 patients was estimated based on an assumed incidence of this outcome in the control group of 20%, an RRR of 30%, an alpha of 5%, a beta of 20%, with a heterogeneity of 11%. A total of 512 patients (37% of the HCRIS) were randomized. The Z-curve crossed the conventional boundaries for *p* = 0.05, but not the monitoring boundaries; HCRIS = heterogeneity-corrected required information size.

**Figure 9 antibiotics-11-01191-f009:**
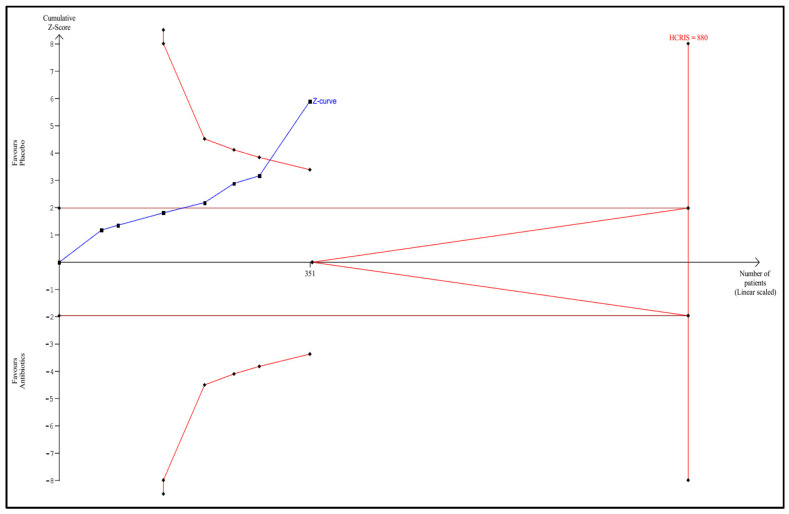
Trial sequential analysis for the length of hospital stay. The HCRIS was 880 patients based on a minimal relevant shortening of hospitalization of 2 days, an empirical variance of 112, and a heterogeneity of 0%. At 351 patients randomized (40% of the HCRIS), the Z-curve crossed both the conventional and sequential monitoring boundaries; HCRIS = heterogeneity-corrected required information size.

## Data Availability

No new data were created or analyzed in this study. Data sharing is not applicable to this article.

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
