# Peer review of "Systematic Review with Trial Sequential Analysis of Prophylactic Antibiotics for Acute Pancreatitis"

_antibiotics, 2022, doi:10.3390/antibiotics11091191_

Round 1

Reviewer 1 Report

Dear Author,

Congratulation for your work. I have briefly reviewed the submitted manuscript on Prophylactic antibiotics in Acute pancreatitis. Following queries needs to be addressed:

Major Comments:

Introduction: Authors started the description with past available studies. I recommend to provide initial background on the topic in the first paragraph. Then, you can switch to the evidence available till date. So, redrafting needs to be done. 

Page 2nd, line 51. no need to mention what meta analysis is used for. Researchers or readers knows the value of meta-analysis. So remove all the statement:  Meta-analyses can increase the power and precision  of an estimated intervention effect (13). However, repeated significance testing within a 52 meta-analyis can also lead to random errors and generate spurious significant results (13). 53 Every trial that is added in a cummulative meta-analysis is regarded as an interim meta- 54 analysis ). If the cummulative result represented by the Z-curve crosses the constructed  monitoring boundaries, a sufficient level of significance is reached and no further research  may be needed. If the Z-curve does not cross the boundary, then evidence is  insufficient in order to reach a conclusion. Since no such analysis has yet been performed,  we intended to quantify the reliability of our data and estimate wheather further research  is still needed.

So, overall, the necessary basic information required is lacking. Provide the background in one paragraph. 

Method: Authors have provided method section at the end of the article. It is difficult to review if all the sections are not in proper flow. Keep method and material after introduction. 

Results: It can be described in concise version. Already, information provided in the figure or tables. there is no need to write in text part. Make it in a concise version. 

Please spelled out RRR in results. 

Discussion is well written. No further action is needed. 

Overall, the article was well written. Images are clear. Conclusion is consistent with the evidence presented.

Author Response

Major Comments:

Introduction: Authors started the description with past available studies. I recommend to provide initial background on the topic in the first paragraph. Then, you can switch to the evidence available till date. So, redrafting needs to be done. 

Page 2nd, line 51. no need to mention what meta analysis is used for. Researchers or readers knows the value of meta-analysis. So remove all the statement:  Meta-analyses can increase the power and precision  of an estimated intervention effect (13). However, repeated significance testing within a 52 meta-analyis can also lead to random errors and generate spurious significant results (13). 53 Every trial that is added in a cummulative meta-analysis is regarded as an interim meta- 54 analysis ). If the cummulative result represented by the Z-curve crosses the constructed  monitoring boundaries, a sufficient level of significance is reached and no further research  may be needed. If the Z-curve does not cross the boundary, then evidence is  insufficient in order to reach a conclusion. Since no such analysis has yet been performed,  we intended to quantify the reliability of our data and estimate wheather further research  is still needed.

So, overall, the necessary basic information required is lacking. Provide the background in one paragraph. 

  • We accepted your suggestions and made the changes accordingly.

Method: Authors have provided method section at the end of the article. It is difficult to review if all the sections are not in proper flow. Keep method and material after introduction. 

  • The ‘Methods’ section was put at the end of the article due to the provided journal template that showed and required it to be put in that “unusual” order. However, we moved it after the ‘Introduction’ section, as it is usual and suggested by Yourself as well, for a better flow.

Results: It can be described in concise version. Already, information provided in the figure or tables. there is no need to write in text part. Make it in a concise version. 

  • We already have nine figures in the article. In order to avoid even more, part of the results have been presented in text format, but those outcomes presented by figures were written in a more concise manner, according to your suggestions.

Please spelled out RRR in results. 

  • We corrected the introduction of the abbreviation RRR and spelled it in full in the ‘Methods’ section, subsection 4.5. Trial Sequential Analysis.

Discussion is well written. No further action is needed. 

  • Thank you.

Overall, the article was well written. Images are clear. Conclusion is consistent with the evidence presented.

Reviewer 2 Report

The author compares various studies done so far for the prophylactic antibiotics against acute pancreatitis by Trial Sequential Analysis (TSA) methodology. TSA is a robust approach for interpreting meta analysis results in providing more control on the type I & II errors.

Introduction:

  1. Introduction line 41, refrain from use of the word fortunately.
  2. Line 50, typo error -methodology that combines…
  3. References 24 & 31 should be cited in introduction (lines 49-50, 53-54).

Result:

1. The title mentions trial sequential analysis for AP, but the meta analysis of references included SAP and AP together in the analysis irrespective o the etiology.

2. Line 62 & 63, the results shown is for total of 3362 references out of which 2806 remained, whereas Fig. 1 it shows 5170 and 4901 respectively. Either correct the numbers shown in Fig 1 or the results. 

3. Please elaborate the figure legends in Figs. 1, 2, 3 and 4.

4. In Figure 2, the data set are inconsistent. The data shown in results doesn’t match what is shown in the figure. For instance, figure shows 9 studies which is 8 in result section, 81 out of 266 patients received PAB according to the figure which is 70 out of 233, even the number of controls. Please include the correct numbers in the figure (line 69 & 70).

5. Line 84 shows 8 trials, whereas in the figure only 7 references are shown.

6. Fig. 9 Los is 880 which is 808 in the results (line117).

7. In the TSA diagrams, maximally allowed type I and II error should be used once and thereafter as alpha and beta.

8. Figure 9 is mislabelled as 8 ( line 172). 

Discussion 

Discussion is well written stating the strengths and limitations of the study reviewed. However, the study could be more refined in terms of etiology and severity of AP.

Materials and methods

Under TSA, with HCRIS, use either alpha and beta or type I and type II errors, but not mixed.

Author Response

The author compares various studies done so far for the prophylactic antibiotics against acute pancreatitis by Trial Sequential Analysis (TSA) methodology. TSA is a robust approach for interpreting meta analysis results in providing more control on the type I & II errors.

Introduction:

  1. Introduction line 41, refrain from use of the word fortunately.
  • This has been corrected accordingly.

  1. Line 50, typo error -methodology that combines…
  • This has been corrected accordingly.

  1. References 24 & 31 should be cited in introduction (lines 49-50, 53-54).
  • The introduction part was redrafted according to suggestions of Reviewer 1. Please see the redrafted version, thank you.

Result:

  1. The title mentions trial sequential analysis for AP, but the meta analysis of references included SAP and AP together in the analysis irrespective o the etiology.

- TSA was performed with the intent to minimize random error and avoid erroneous conclusions with probable overestimation of treatment effect, since we were aware of many published trials being conducted on a very limited number of patients with inadequate or non-existent sample size calculation. The identified studies included different patients according to etiology, and many included studies did not even report on specific etiology, as it is shown in Supplementary Table 1. Therefore, it was impossible for us to perform subgroup analysis according to etiology, since we could acquire raw data for patients with a specific etiology, and performing meta-analysis only with those studies specifying etiology and giving patients data according to etiology would reduce the number of patients even more, and therefore reduce the objectivity and interpretability of such an analysis furthermore.

- Separate analysis of patients with severe and non-severe forms of AP was not possible due to several reasons. First, certain identified studies did not report at all on severity of included patients. Second, some studies reported on the percentage of patients having severe forms of AP but did not give specific data for this subgroup of patients. Third, the definition of severity changed substantially through the years, from defining severe forms just by presence of necrosis, across different scoring systems (APACHE II ≥ 8; Randon ≥ 3, etc), to the current definition based on presence of persistent organ failure according to the revised Atlanta criteria. Such different criteria and definitions would generate a very heterogenous population in which severe patients would very likely include also moderately severe and mild AP. These are the reason why such analysis would not be feasible, and that’s why we kept a general approach by keeping all AP patients and did not emphasize and specify severe patients. We definitely agree that effects of antibiotics would be much more of interest in clearly severe and critically-ill AP patients, however available evidence does not allow such analysis and interpretations.

  1. Line 62 & 63, the results shown is for total of 3362 references out of which 2806 remained, whereas Fig. 1 it shows 5170 and 4901 respectively. Either correct the numbers shown in Fig 1 or the results. 

- This was corrected according to data given in Figure 1, which were the updated and correct ones.

  1. Please elaborate the figure legends in Figs. 1, 2, 3 and 4.

- We elaborated the legends for Figures 2, 3 and 4, and reduced the corresponding text in the Results section. We left the legend for Figure 1 in its current form, since the search and selection of studies is described in detail in the text, and we don’t see the need for further explanation.

  1. In Figure 2, the data set are inconsistent. The data shown in results doesn’t match what is shown in the figure. For instance, figure shows 9 studies which is 8 in result section, 81 out of 266 patients received PAB according to the figure which is 70 out of 233, even the number of controls. Please include the correct numbers in the figure (line 69 & 70).

- data given in text have been corrected according to the data given in Figure 2.

  1. Line 84 shows 8 trials, whereas in the figure only 7 references are shown.

- we could not identify to which figure and corresponding text in line 84 were you referring to?

  1. Fig. 9 Los is 880 which is 808 in the results (line117).

- this has been corrected accordingly.

  1. In the TSA diagrams, maximally allowed type I and II error should be used once and thereafter as alpha and beta.

- this has been corrected accordingly.

  1. Figure 9 is mislabelled as 8 ( line 172). 

- this has been corrected accordingly.

Discussion 

Discussion is well written stating the strengths and limitations of the study reviewed. However, the study could be more refined in terms of etiology and severity of AP.

Materials and methods

Under TSA, with HCRIS, use either alpha and beta or type I and type II errors, but not mixed.

- this has been corrected accordingly

Reviewer 3 Report

The abbreviations should be explained at the first time when they appear in the article. Some of them are missing, for example: the HCRIS appears for the first time in point 2.1 and its explanation can be found in point 4.5 or under the figures (starting from number 5). The same thing is for the UTI (line 111). There are some intervals between words missing (line 63 and some others).

The abstract doesn’t meet the article. In the results section of the abstract there is a statement that eighteen trials with 1134 patients were included. However, the article tells us that there were 21 articles with 1383 patients included.

The impact of antibiotics on the course of acute pancreatitis was largely discussed in many researches. In my opinion, researches from the following years 1976, 1995, 1993, 1996, 1997 should be excluded. The progress in medicine (especially in diagnostics – widely used CT, different criteria for the course of acute pancreatitis) is tremendous, therefore, including control trials from that period may lead to errors in judgement.

Author Response

The abbreviations should be explained at the first time when they appear in the article. Some of them are missing, for example: the HCRIS appears for the first time in point 2.1 and its explanation can be found in point 4.5 or under the figures (starting from number 5). The same thing is for the UTI (line 111). There are some intervals between words missing (line 63 and some others).

  • The journal template required the Methods section to be put at the end. By moving it after the Introduction part, and by introducing the abbreviations properly, we achieved a better flow of the article.

The abstract doesn’t meet the article. In the results section of the abstract there is a statement that eighteen trials with 1134 patients were included. However, the article tells us that there were 21 articles with 1383 patients included.

  • This has been corrected according to the main text of the article.

The impact of antibiotics on the course of acute pancreatitis was largely discussed in many researches. In my opinion, researches from the following years 1976, 1995, 1993, 1996, 1997 should be excluded. The progress in medicine (especially in diagnostics – widely used CT, different criteria for the course of acute pancreatitis) is tremendous, therefore, including control trials from that period may lead to errors in judgement.

  • We ran and provided (below) results from meta-analyses with trials published before 2000 being excluded. The main results did not change significantly. We noticed that by excluding these trials, there is no significant reduction of urinary tract infection rate. These small trials which were conducted and published more than 20 years ago were also our main concern regarding the potential random and systematic errors leading to erroneous conclusions and judgment, and that is why we decided to perform TSA, which actually confirmed there is no evidence of antibiotics having any beneficial effect on reducing infected pancreatic necrosis and mortality, and that the evidence for prophylactic use against other infections in this specific group of patients is not solid enough. Since such analysis led to no new and different results, in order to reduce the amount of text, we have not included and presented the analyses below.

  • Mortality – RR 1.03 (95% CI 0.76 to 1.39), I2=0%.
  • All infections – RR 0.63 (95% CI 0.47 to 0.84), I2=40%
  • Organ failure – RR 0.83 (95% CI 0.65 to 1.05), I2=0%
  • Pneumonia – RR 0.81 (95% CI 0.46 to 1.42). I2=0%
  • Infected pancreatic necrosis – RR 0.89 (95% CI 0.66 to 1.20), I2=0%
  • Sepsis – RR 0.52 (95% CI 0.27 to 1.00), I2=0%
  • UTI – RR 0.77 (95% CI 0.35 to 1.70), I2=0%
  • Acute renal failure – RR 0.94 (95% CI 0.53 to 1.69), I2=0%
  • Acute respiratory failure – RR 0.83 (95% CI 0.53 to 1.30), I2=0%
  • Cardiovascular failure – RR 0.89 (95% CI 0.38 to 2.11), I2=0%
  • Length of hospitalization – MD -6.60 (95% CI -8.86 to -4.35), I2=0%

Round 2

Reviewer 3 Report

The manuscript has been improved.